# Strong Preferences Affect the Robustness of Preference Models and Value Alignment

**Ziwei Xu**
Department of Computer Science
National University of Singapore
ziwei.xu@u.nus.edu

**Mohan Kankanhalli**
Department of Computer Science
National University of Singapore
mohan@comp.nus.edu.sg

## Abstract

Value alignment, which aims to ensure that large language models (LLMs) and other AI agents behave in accordance with human values, is critical for ensuring safety and trustworthiness of these systems. A key component of value alignment is the modeling of human preferences as a representation of human values. In this paper, we investigate the robustness of value alignment by examining the sensitivity of preference models. Specifically, we ask: how do changes in the probabilities of some preferences affect the predictions of these models for other preferences? To answer this question, we theoretically analyze the robustness of widely used preference models by examining their sensitivities to minor changes in preferences they model. Our findings reveal that, in the Bradley-Terry and the Placket-Luce model, the probability of a preference can change significantly as other preferences change, especially when these preferences are dominant (i.e., with probabilities near 0 or 1). We identify specific conditions where this sensitivity becomes significant for these models and discuss the practical implications for the robustness and safety of value alignment in AI systems.

## 1 Introduction

Value alignment (Gabriel & Ghazavi, 2021) aims to ensure that AI agents, such as large language models (LLMs) (OpenAI, 2023; Llama Team, 2024), behave in accordance with human values. It is critical for ensuring safety and trustworthiness of AI systems. An important component of value alignment is the modeling of preferences, where preferences of individual or groups of people (e.g., citizens of a country) are collected as samples of choices made by the people over options given contexts of decisions and are then fit by probabilistic frameworks to predict preferences in unseen contexts. For example, in the Reinforcement Learning from Human Feedback (RLHF) framework, alternative model outputs for the same prompt are shown to the human subjects in order to elicit their preferences. A comprehensive set of such preferences are then used to train a reward/preference model, which is subsequently used to train a target agent via proximal policy optimization (PPO) (Schulman et al., 2017). Recent value alignment approaches, such as direct policy optimization (DPO) (Rafailov et al., 2023), uses reparameterized implicit preference models in optimization.

One of the most widely used preference model in value alignment research is the Bradley-Terry model (Bradley & Terry, 1952), which is the pairwise case for the more general Plackett-Luce model (Plackett, 1975; Luce, 1959). The study on these for value alignment has been focused on how to better fit a fixed target distribution of preferences (Rafailov et al., 2023; Azar et al., 2024; Xu et al., 2024). However, there has been limited exploration of how changes in the probability of a modelled preference could influence others within these models. As will be shown later in Example 1, given options $\{o_i, o_j, o_k\}$, a Bradley-Terry model could assign $p^{\text{BT}}(o_i \succ o_j)$ very differently, when there is slight change in $p^{\text{BT}}(o_i \succ o_k)$ and $p^{\text{BT}}(o_k \succ o_j)$. Understanding these relationships is crucial for assessing the robustness of preference models in dynamic or uncertain environments, particularly when observed preferences may shift due to noise, randomness in optimization, or evolving conditions in the dataset, and could impact the robustness and safety of value alignment for AI systems (Anwar et al., 2024) This is especially relevant when preference models are sensitive to these shifts, as such sensitivity may lead to significant changes in the probabilities of other unseen preferences.

This paper explores the robustness of value alignment by examining the sensitivity in preference models. We ask the core question: *under common preference models, how does a change in the probabilities of some preferences influences the model's predictions for other preferences?* To answer this question, we begin by outlining the definitions and assumptions of a preference model in Section 2.1. Next, we explore a general pairwise model in Section 2.2 and show that the probability of any given preference can be expressed as a function of other preference probabilities. By analyzing this function, we show that probabilities of pairwise preferences may exhibit sensitivity to changes in other preferences, particularly when those preferences approach dominance, which is marked by probabilities near 0 or 1. In Section 2.3, we then examine the Bradley-Terry model as a special pairwise preference model and identify scenarios where such sensitivity arises. Lastly, we extend our discussion to the $K$-tuple Plackett-Luce model in Section 2.4.

The contributions of this paper are three-fold. First, we present a theoretical analysis that reveals the relation between preference probabilities in general pairwise preference models and the $K$-tuple Plackett-Luce model. Second, we show that in all the models studied in this paper, the probability of a given preference can exhibit sensitivity to changes in other preference probabilities as preferences approach dominance, potentially compromising robustness of value alignment processes. Furthermore, we identify the exact conditions under which this sensitivity occurs for the Plackett-Luce model (and the Bradley-Terry model). Third, we discuss the practical implications of our theoretical analysis for the robustness and safety of value alignment in AI systems.

## 2 ANALYSIS OF PREFERENCE MODELS

We consider the following value alignment setting in our analysis. Suppose there is a group of human subjects who share certain values which we hope to align target AI agents with. In a decision-making context, the values guide the evaluation of the strengths of candidate options, producing a ranking of these options. A preference is thus a ranking over candidate options, denoted as $\mathcal{O} = \{o_i \mid i = 1, 2, \ldots, N\}$, based on the their corresponding strengths, represented as scores $\mathcal{S} = \{s_i \mid s_i \in \mathbb{R}, i = 1, 2, \ldots, N\}$. Since it is difficult to directly obtain the scores, we instead query the subjects' preferences and fit models to estimate the scores of the options. The model then captures the group's values by learning and predicting the probability of the subjects expressing any given preference.

### 2.1 DEFINITIONS

Formally, we define a $K$-tuple preference as a ranking of $K$ options:

**Definition 1.** ($K$-tuple preference.) A $K$-tuple preference is a ranking $o_{\omega_1} \succ o_{\omega_2} \succ \ldots \succ o_{\omega_{K-1}} \succ o_{\omega_K}$ over $K$ items in $\mathcal{O}$. It can equivalently expressed as a $K$-permutation of $\mathcal{O}$, or $\boldsymbol{\omega} = \left(o_{\omega_1}, o_{\omega_2}, \ldots, o_{\omega_{K-1}}, o_{\omega_K}\right)$, where $1 \leq \omega_* \leq N$.

The probability of a particular preference $\boldsymbol{\omega}$ (being expressed by a certain group of subjects), denoted $p_{\boldsymbol{\omega}}$, is predicted by a preference model.

**Definition 2.** ($K$-tuple preference model.) A $K$-tuple preference model is a function $f$ that predicts the probability of a given $K$-tuple preference. Formally, $p_{\boldsymbol{\omega}} = f(\boldsymbol{\omega})$, where $f : \text{Perm}(\mathcal{O}, K) \to (0, 1)$, $\text{Perm}(\mathcal{O}, K)$ represents the set of all $K$-permutations of set $\mathcal{O}$, and $\boldsymbol{\omega} \in \text{Perm}(\mathcal{O}, K)$.

We will use superscripts to indicate the specific preference model, and subscripts are used to denote the corresponding preference. For example, $p_{ij}^{\text{BT}}$ refers to the probability of $(o_i, o_j)$ under the Bradley-Terry model, and $p_{\boldsymbol{\omega}}^{(K)}$ to the probability of $\boldsymbol{\omega}$ under a $K$-tuple Plackett-Luce model.

**Assumption 1.** (Preference models depend only on score differences.) Following assumptions in traditional preference models (Thurstone, 1927; Bradley & Terry, 1952; Plackett, 1975; Luce, 1959), we assume that a preference model depends only on the differences between the scores of the options. Formally, $f(\boldsymbol{\omega}) = g(s_{\omega_1} - s_{\omega_2}, s_{\omega_1} - s_{\omega_3}, \ldots, s_{\omega_1} - s_{\omega_K}, s_{\omega_2} - s_{\omega_3}, \ldots, s_{\omega_{K-1}} - s_{\omega_K})$, where $s_{\omega_i}$ denotes the score of option $\omega_i$.

We study the robustness of value alignment by analyzing the sensitivity of the probability of a preference w.r.t. other preferences. Intuitively, higher sensitivity means lower robustness because in this case the probability of a particular preference will fluctuate significantly with minor changes in the probability of other preferences. Below we define a function's sensitivity as the rate of change

w.r.t. its arguments. Specifically, we use $M$-sensitivity to describe situations where the rate of change exceeds $M$.

**Definition 3.** ($M$-sensitivity.) Let $h$ be a function defined over $\boldsymbol{x} = (x_1, x_2, \ldots, x_L)$, where $L \in \mathbb{N}$ is the number of arguments of $h$. Then $h(\boldsymbol{x})$ is $M$-sensitive to $x_i$ at point $\boldsymbol{x}'$ if $|\partial h(\boldsymbol{x})/\partial x_i|_{\boldsymbol{x}=\boldsymbol{x}'}| > M$ and $1 \le i \le L$, where $M > 0$.

We are also interested in regions where $h$ is $M$-sensitive, which is defined as follows.

**Definition 4.** ($M$-sensitive region.) Let $\mathrm{Dom}(h)$ represent the domain of $h(\boldsymbol{x})$. The $M$-sensitive region of $h(\boldsymbol{x})$ w.r.t. $x_i$ is $\Omega_M(h, x_i) : \{\boldsymbol{x}' \in \mathrm{Dom}(h) : |\partial h/\partial x_i|_{\boldsymbol{x}=\boldsymbol{x}'}| > M\}$.

Intuitively, we use $A\left(\Omega_M(h, x_i)\right) = \int_{\Omega_M(h, x_i)} 1 \mathrm{d}\boldsymbol{x}$, the area of the region defined by $\Omega_M(h, x_i)$, to quantify the extent to which $h(x)$ is affect by its $M$-sensitivity w.r.t. $x_i$. In the following sections, we will focus on $M$-sensitivity with $M > 1$, where the function changes faster than its arguments.

## 2.2 ANALYSIS OF A GENERAL PAIRWISE PREFERENCE MODEL

This section discusses the probability of a pair of options $(o_i, o_j)$ as $p_{ij}^{(2)} = g(s_i - s_j)$, where $g$ is a preference model that predicts the probability $p_{ij}$ of the ordered pair (i.e., 2-tuple) $(o_i, o_j)$ based on the score difference $s_i - s_j \in \mathbb{R}$. A common example is the Bradley-Terry model $p_{ij}^{\mathrm{BT}} = 1/\left(1 + \exp\left(-(s_i - s_j)\right)\right)$. Without restricting the discussion to any specific preference model, we assume that $g(s_i - s_j)$ satisfies the following properties:

**Assumption 2.** (Strictly increasing.) $g$ is a strictly increasing function. Therefore, $p_{ij}^{(2)}$ grows with $s_i - s_j$. In other words, $o_i$ is more preferred over $o_j$ whenever $s_i - s_j$ grows.

**Assumption 3.** (Limits at infinity.) $\lim_{x \to -\infty} g(x) = 0$ and $\lim_{x \to +\infty} g(x) = 1$. Therefore, $p_{ij}^{(2)}$ is bounded within $(0, 1)$, and when $(s_i - s_j)$ goes to positive (negative) infinity, $p_{ij}^{(2)} = 1$ (or 0).

**Assumption 4.** (Symmetry.) $\forall x \in \mathbb{R}, g(x) + g(-x) = 1$. Therefore, $\forall o_i, o_j \in \mathcal{O}, p_{ij}^{(2)} + p_{ji}^{(2)} = g(s_i - s_j) + g(s_j - s_i) = 1$, meaning a higher preference for $o_i$ implies a lower preference for $o_j$.

**Assumption 5.** (Continuous differentiability.) $g(x)$ is continuously differentiable. Therefore, $g$ and its derivative are reasonably smooth.

The following lemma says that, in a pairwise preference model, the probability of any given preference can be expressed as a function of two other preference probabilities.

**Lemma 1.** For all $o_i, o_j, o_k \in \mathcal{O}$, and under the pairwise model $p_{ij}^{(2)} = g(s_i - s_j)$ following assumptions above,

$$p_{ij}^{(2)} = g\left(g^{-1}(p_{ik}^{(2)}) + g^{-1}(p_{kj}^{(2)})\right), \tag{1}$$

where $g^{-1} : (0, 1) \to \mathbb{R}$ is the inverse of $g$, mapping a probability to a difference of scores.

*Proof.* $p_{ij}^{(2)} = g(s_i - s_j) = g\left((s_i - s_k) + (s_k - s_j)\right) = g\left(g^{-1}(p_{ik}^{(2)}) + g^{-1}(p_{kj}^{(2)})\right).$ $\qquad\square$

Lemma 1 is the basis of analysis below as we study $p_{ij}^{(2)}$ as a function of $p_{ik}^{(2)}$ and $p_{kj}^{(2)}$. It suggests that under a pairwise preference model, changes in some probabilities (e.g., $p_{ik}^{(2)}$ and/or $p_{kj}^{(2)}$) leads to changes in other probabilities (e.g., $p_{ij}^{(2)}$). We now study how significant such changes could be by focusing on the sensitivity of $p_{ij}^{(2)}$ to $p_{ik}^{(2)}$ and $p_{kj}^{(2)}$. Towards this, we first examine $g'$ and $g^{-1}$, the derivative and the inverse of $g$, as they are the key components of Eq. (1).

We first consider $g'$ and show that it must approach $0$ at infinity.

**Lemma 2.** Let $g'(x) = \frac{\mathrm{d}g(x)}{\mathrm{d}x}$, then $\lim_{x \to -\infty} g'(x) = \lim_{x \to +\infty} g'(x) = 0$.

*Proof.* We prove $\lim_{x \to +\infty} g'(x) = 0$ by contradiction. Proof for $\lim_{x \to -\infty} g'(x) = 0$ is similar.

Suppose $\lim_{x \to +\infty} g'(x) \ne 0$. Note that by Assumption 2, for all $x \in \mathbb{R}, g'(x) \ge 0$. Then $\exists \varepsilon_0 > 0$ such that $\forall M > 0, \exists x_0 > M$ with $g'(x_0) \ge \varepsilon_0$. Now consider the neighborhood of $x_0$. By

Assumption 5, $g'(x)$ is a continuous function. Since $g'(x_0) \geq \varepsilon_0$, there exists $\delta_0 > 0$ such that for all $x \in [x_0, x_0 + \delta]$, $g'(x) \geq \varepsilon_0/2$. As a result, $g(x_0 + \delta_0) - g(x_0) = \int_{x_0}^{x_0+\delta_0} g'(x)\mathrm{d}x \geq \delta_0 \frac{\varepsilon_0}{2}$, hence $g(x_0 + \delta_0) \geq g(x_0) + \delta_0 \frac{\varepsilon_0}{2}$.

Now take $x_0 \to +\infty$ (as $M$ goes to infinity). By Assumption 3, $\lim_{x \to +\infty} g(x) = 1$. Therefore $\lim_{x_0 \to +\infty} g(x_0 + \delta_0) \geq 1 + \delta_0 \frac{\varepsilon_0}{2}$. This implies $\lim_{x \to +\infty} g(x) > 1$ and contradicts with Assumption 3. Therefore the converse of our assumption, i.e., $\lim_{x \to +\infty} g'(x) = 0$, must be true. □

We further examine $g'$ and show the following two lemmas about its monotonicity and symmetry.

**Lemma 3.** Let $g^{-1}$ be the inverse of $g$, then $g^{-1}(x)$ is strictly increasing.

*Proof.* Let $u = g(x_1)$ and $v = g(x_2)$, where $x_1 < x_2$. Then $u < v \Leftrightarrow g(x_1) < g(x_2) \Leftrightarrow x_1 < x_2 \Leftrightarrow g^{-1}(u) < g^{-1}(v)$. Therefore, $g^{-1}$ is strictly increasing. □

**Lemma 4.** $\forall 0 < x < 1, g^{-1}(x) + g^{-1}(1-x) = 0$.

*Proof.* Let $u = g^{-1}(x), v = g^{-1}(1-x)$. Then $g(u) = x, g(v) = 1 - x$. By Assumption 2 $g(x) + g(-x) = 1$. Substituting $x$ with $u$, we get $g(u) + g(-u) = 1 \Leftrightarrow x + g(-u) = 1 \Leftrightarrow g(-u) = 1 - x = g(v)$. Since $g$ is strictly increasing by Assumption 2, $g(-u) = g(v) \Leftrightarrow -u = v \Leftrightarrow u + v = 0$. Therefore, $g^{-1}(x) + g^{-1}(1-x) = u + v = 0$. □

With all the lemmas above, we are ready to discuss the $M$-sensitivity of $p_{ij}^{(2)}$ w.r.t. $p_{ik}^{(2)}$. In fact, we can show that such sensitivity exists for all $M > 0$, as follows.

**Theorem 1.** For all $M > 0$, there exists $0 < p_0, p_{kj}^{(2)} < 1$ such that for all $p_0 < p_{ik}^{(2)} < 1$, $p_{ij}^{(2)}$ is $M$-sensitive to $p_{ik}^{(2)}$.

*Proof.* We prove this by showing that, for all $M > 0$, there exists $0 < p_0, p_{kj}^{(2)} < 1$, such that $\left| \frac{\partial p_{ij}^{(2)}}{\partial p_{ik}^{(2)}} \right| > M$ for all $p_0^{(2)} < p_{ik}^{(2)} < 1$. Due to Lemma 1, $p_{ij}^{(2)} = g\left(g^{-1}(p_{ik}^{(2)}) + g^{-1}(p_{kj}^{(2)})\right)$. Taking the partial derivative of $p_{ij}^{(2)}$ w.r.t. $p_{ik}^{(2)}$, by the chain rule we have

$$\frac{\partial p_{ij}^{(2)}}{\partial p_{ik}^{(2)}} = \left.\frac{\mathrm{d}g}{\mathrm{d}x}\right|_{x=g^{-1}(p_{ik}^{(2)})+g^{-1}(p_{kj}^{(2)})} \times \frac{\mathrm{d}g^{-1}}{\mathrm{d}p_{ik}^{(2)}} = \left.\frac{\mathrm{d}g}{\mathrm{d}x}\right|_{x=g^{-1}(p_{ik}^{(2)})+g^{-1}(p_{kj}^{(2)})} \times \left.\left(\frac{\mathrm{d}g}{\mathrm{d}x'}\right)^{-1}\right|_{x'=g^{-1}(p_{ik}^{(2)})}$$

$$= g'\left(g^{-1}(p_{ik}^{(2)}) + g^{-1}(p_{kj}^{(2)})\right) \cdot \frac{1}{g'\left(g^{-1}(p_{ik}^{(2)})\right)} = A_1 \cdot A_2. \tag{2}$$

Since $g(x)$ is strictly increasing on $\mathbb{R}$, there exists $\delta > 0$ such that $g'(\delta) > 0$.

Let $\epsilon = \frac{g'(\delta)}{M} > 0$. Due to Lemma 2, $\lim_{x \to +\infty} g'(x) = 0$. Therefore, $\exists x_0 \in \mathbb{R}$ s.t. $\forall x > x_0$, $g'(x) < \epsilon = \frac{g'(\delta)}{M}$, or equivalently $\frac{1}{g'(x)} > \frac{M}{g'(\delta)}$. Consider the $A_2$ part of Eq. (2). Let $p_0 = g(x_0)$ and consider all $p_{ik}^{(2)} > p_0$. Since $g^{-1}$ is strictly increasing (Lemma 3), $g^{-1}(p_{ik}^{(2)}) > g^{-1}(p_0) = x_0$. Therefore, $\forall p_{ik}^{(2)} > p_0$, $A_2 = \frac{1}{g'\left(g^{-1}(p_{ik}^{(2)})\right)} > \frac{M}{g'(\delta)}$.

Consider the $A_1$ part of Eq. (2). Let $p_{kj}^{(2)} = g\left(g^{-1}\left(1 - p_{ik}^{(2)}\right) + \delta\right)$. Due to Lemma 4: $A_1 = g'\left(g^{-1}(p_{ik}^{(2)}) + g^{-1}(p_{kj}^{(2)})\right) = g'\left(g^{-1}(p_{ik}^{(2)}) + g^{-1}(1 - p_{ik}^{(2)}) + \delta\right) = g'(\delta)$.

Therefore, there exists $\delta > 0$ and $x_0 \in \mathbb{R}$, such that $\forall p_{ij}^{(2)} > p_0 = g(x_0)$, and $p_{kj}^{(2)} = g\left(g^{-1}\left(1 - p_{ik}^{(2)}\right) + \delta\right)$, $\left| \frac{\partial p_{ij}^{(2)}}{\partial p_{ik}^{(2)}} \right| = A_1 \cdot A_2 > g'(\delta)\frac{M}{g'(\delta)} = M$. □

Theorem 1 suggests that $p_{ij}^{(2)}$ could be $M$-sensitive to $p_{ik}^{(2)}$ for any $M$, regardless of how large $M$ is. Specifically, the proof demonstrates that this sensitivity arises when $p_{ik}^{(2)}$ approaches a sufficiently large value (close to 1). Note that the proof only considers the case when $x \to +\infty$ when applying

Lemma 2. In fact, it can be shown that similar sensitivity arises when $p_{ik}^{(2)}$ is sufficiently small. To conclude, $p_{ij}^{(2)}$ is sensitive to $p_{ik}^{(2)}$ when it approaches dominance marked by probabilities near 0 or 1.

The definition of general pairwise model is too abstract for us to gain more intuitive understanding of the sensitivity demonstrated in Theorem 1. For example, one may want to know the exact sensitivity of $p_{ij}^{(2)}$ w.r.t. a given $p_{ik}^{(2)}$. It may also be interesting to compare the sensitivity of different preference models. In the following sections, we study Bradley-Terry model as a concrete pairwise model for its sensitivity, and extend our discussion to the $K$-tuple Plackett-Luce models.

## 2.3 ANALYSIS ON THE BRADLEY-TERRY MODEL

The Bradley-Terry model is a widely-used special case of the general pairwise preference model discussed in the section above. Under this model, $p_{ij}^{\mathrm{BT}} = g_{\mathrm{BT}}(s_i - s_j) = 1/\left(1 + \exp\left(-\left(s_i - s_j\right)\right)\right)$. According to Lemma 1, we write $p_{ij}^{\mathrm{BT}}$ as a function of $p_{ik}^{\mathrm{BT}}, p_{kj}^{\mathrm{BT}}$:

$$p_{ij}^{\mathrm{BT}} = \frac{1}{1 + \left(1 - p_{ik}^{\mathrm{BT}}\right)\left(1 - p_{kj}^{\mathrm{BT}}\right)/p_{ik}^{\mathrm{BT}}/p_{kj}^{\mathrm{BT}}}. \tag{3}$$

$\partial p_{ij}^{\mathrm{BT}}/\partial p_{ik}^{\mathrm{BT}}$ can be derived as

$$\frac{\partial p_{ij}^{\mathrm{BT}}}{\partial p_{ik}^{\mathrm{BT}}} = \frac{p_{kj}^{\mathrm{BT}}\left(1 - p_{kj}^{\mathrm{BT}}\right)}{\left(p_{ik}^{\mathrm{BT}} + p_{kj}^{\mathrm{BT}} - 2p_{ik}^{\mathrm{BT}}p_{kj}^{\mathrm{BT}} - 1\right)^2}. \tag{4}$$

**When is Bradley-Terry model sensitive?** Consider the $M$-sensitivity of $p_{ij}^{\mathrm{BT}}$ w.r.t. $p_{ik}^{\mathrm{BT}}$ by solving $\left|\partial p_{ij}^{\mathrm{BT}}/\partial p_{ik}^{\mathrm{BT}}\right| > M$. It turns out that, for $M > 1$, $\Omega_M(p_{ij}^{\mathrm{BT}}, p_{ik}^{\mathrm{BT}})$ consists of two regions:

Case 1: $0 < p_{kj}^{\mathrm{BT}} < \dfrac{1}{1 + M}$, $\gamma_0 < p_{ik}^{\mathrm{BT}} < 1$,

Case 2: $1 - \dfrac{1}{1 + M} < p_{kj}^{\mathrm{BT}} < 1$, $0 < p_{ik}^{\mathrm{BT}} < \gamma_0$, where $\gamma_0 = 1 - \dfrac{\sqrt{\frac{\left(1/p_{kj}^{\mathrm{BT}} - 1\right)}{M}} - 1}{1/p_{kj}^{\mathrm{BT}} - 2}$. $\tag{5}$

We can compute the area of the regions above, which gives the following corollary.

**Proposition 1.** The area of $\Omega_M\left(p_{ij}^{\mathrm{BT}}, p_{ik}^{\mathrm{BT}}\right)$, where $M > 1$, is

$$A\left(\Omega_M\left(p_{ij}^{\mathrm{BT}}, p_{ik}^{\mathrm{BT}}\right)\right) = \frac{1}{2}\ln\left(\frac{M - 1}{M + 1}\right) + \frac{1}{2\sqrt{M}}\ln\left(\frac{\sqrt{M} + 1}{\sqrt{M} - 1}\right).$$

Readers are referred to Appendix B.1 for a proof of Proposition 1. The analysis above can also be applied to $\Omega_M(p_{ij}^{\mathrm{BT}}, p_{kj}^{\mathrm{BT}})$. Fig. 1 illustrates $\Omega_M(p_{ij}^{\mathrm{BT}}, p_{kj}^{\mathrm{BT}})$ and $\Omega_M(p_{ij}^{\mathrm{BT}}, p_{ik}^{\mathrm{BT}})$. As the figure shows, consistent with our discussion on the general pairwise model, $p_{ij}^{\mathrm{BT}}$ becomes increasingly sensitive to $p_{ik}^{\mathrm{BT}}$ and $p_{kj}^{\mathrm{BT}}$ as they approach 0 or 1. Furthermore, the size of $M$-sensitive region shrinks as $M$ increases. However, these regions are not negligible even for fairly large values of $M$.

Below we present an example demonstrating the possible negative consequences of sensitivities. We discuss further implications of our discussions so far in Section 3.1.

**Example 1.** Suppose $(p_{ik}^{\mathcal{D}}, p_{kj}^{\mathcal{D}}) = (0.99, 0.02)$ are the probabilities specified by a dataset $\mathcal{D}$ of preferences. The dataset is used to train two preference models, $p^{\mathrm{BT,1}}$ and $p^{\mathrm{BT,2}}$, which will be deployed on $(o_i, o_j)$. Suppose that after training, due to factors such as randomness in optimization or differences in hyperparameters, the probabilities of $(o_i, o_k)$ learned by these models deviate by $1\%$ deviation from the training set. Let $(p_{ik}^{\mathrm{BT,1}}, p_{kj}^{\mathrm{BT,1}}) = (0.9999, 0.02)$ and $(p_{ik}^{\mathrm{BT,2}}, p_{kj}^{\mathrm{BT,2}}) = (0.9801, 0.02)$ be the actual probabilities learned by the model. While this difference appears minor, according to Eq. (3), the models will behave very differently for $(o_i, o_j)$, with $p_{ij}^{\mathrm{BT,1}} \approx 0.96$ and $p_{ij}^{\mathrm{BT,2}} \approx 0.50$. Such difference arises because, by Eq. (5), $(p_{ik}^{\mathrm{BT}}, p_{kj}^{\mathrm{BT}}) = (0.99, 0.02)$ belongs to $\Omega_{20}(p_{ij}^{\mathrm{BT}}, p_{ik}^{\mathrm{BT}})$, meaning that 1 unit of deviation in $p_{ik}^{\mathrm{BT}}$ induces more than 20 units of changes in $p_{ij}^{\mathrm{BT}}$.

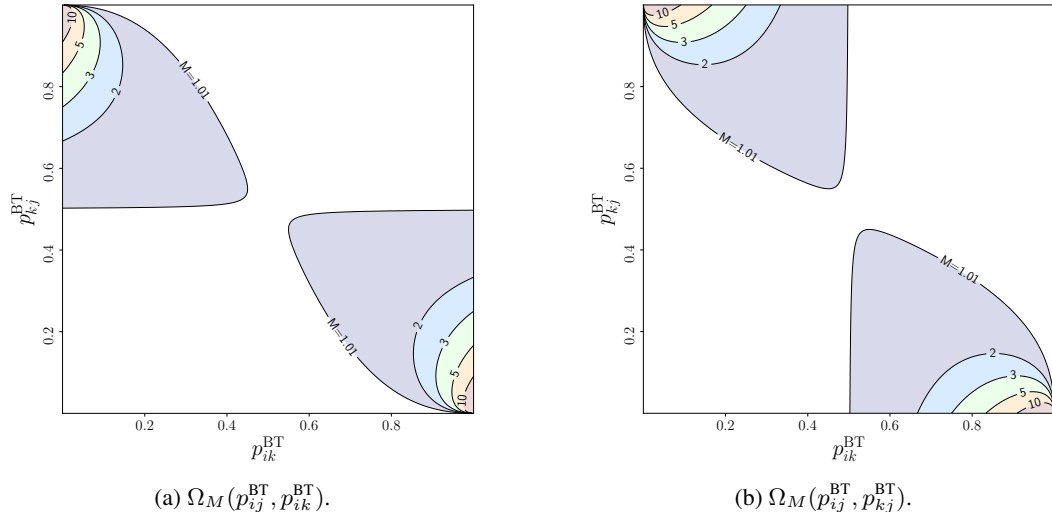

(a) $\Omega_M(p_{ij}^{\mathrm{BT}}, p_{ik}^{\mathrm{BT}})$.

(b) $\Omega_M(p_{ij}^{\mathrm{BT}}, p_{kj}^{\mathrm{BT}})$.

Figure 1: $M$-sensitive regions of $p_{ij}^{\mathrm{BT}}$ w.r.t. $p_{ik}^{\mathrm{BT}}$ and $p_{kj}^{\mathrm{BT}}$, for $M = \{1.01, 2, 3, 5, 10\}$.

## 2.4 EXTENSION TO THE $K$-TUPLE PLACKETT-LUCE MODEL

The Bradley-Terry model can be extended to the more general $K$-tuple Plackett-Luce model, which describes probabilities of preferences over $K$ options, where $2 \leq K \leq N$. In the extended model, a preference is a $K$-tuple $\boldsymbol{\omega} = \left(o_{\omega_1}, o_{\omega_2}, \ldots, o_{\omega_{K-1}}, o_{\omega_K}\right)$, where $o_{\omega_*} \in \mathcal{O}$. The probability of this preference is defined as:

$$p_{\boldsymbol{\omega}}^{(K)} = \prod_{u=1}^{K-1} \frac{\exp\left(s_{w_u}\right)}{\sum_{v=u}^{K} \exp\left(s_{\omega_v}\right)} = \prod_{u=1}^{K-1} \frac{1}{1 + \sum_{v=u+1}^{K} \exp\left(-\left(s_{\omega_u} - s_{\omega_v}\right)\right)}. \tag{6}$$

When $K = 2$, the model degenerates to the Bradley-Terry model.

We are interested in whether sensitivity analysis for Bradley-Terry model can generalize to $K$-tuple preference models for cases where $K > 2$. In particular, we address the following questions: (1) how $p_{\boldsymbol{\omega}}^{(K)}$ can be expressed as a function of other preference probabilities, (2) its $M$-sensitivity w.r.t. these probabilities, and (3) the extent of its sensitivity in comparison with the Bradley-Terry model.

**Lemma 5.** Let $\boldsymbol{\omega}$ be a $K$-tuple preference, where $K > 2$. Let $\boldsymbol{\omega}_{uv} = \left(\boldsymbol{\omega}'_{uv}; o_{\omega_u}, o_{\omega_v}\right)$ be a $K$-permutation of $\mathcal{O}$ with $o_{\omega_u}$ and $o_{\omega_v}$ being the last two elements and $\boldsymbol{\omega}'_{uv}$ being any $(K-2)$-permutation of $\mathcal{O} \setminus \{o_{\omega_u}, o_{\omega_v}\}$. Then $p_{\boldsymbol{\omega}}^{(K)}$ is a function of $p_{\boldsymbol{\omega}_{vu}}^{(K)} / p_{\boldsymbol{\omega}_{uv}}^{(K)}$, where $1 \leq u < v \leq K$. More specifically,

$$p_{\boldsymbol{\omega}}^{(K)} = \prod_{u=1}^{K-1} \frac{1}{1 + \sum_{v=u+1}^{K} \frac{p_{\boldsymbol{\omega}_{vu}}^{(K)}}{p_{\boldsymbol{\omega}_{uv}}^{(K)}}}. \tag{7}$$

*Proof.* Note that there are $(K-1)$ entries in the product of Eq. (6). Moreover, $p_{\boldsymbol{\omega}_{uv}}^{(K)}$ and $p_{\boldsymbol{\omega}_{vu}}^{(K)}$ share the same first $(K-2)$ entries because $\boldsymbol{\omega}_{uv}$ and $\boldsymbol{\omega}_{vu}$ share the same prefix $\boldsymbol{\omega}'_{uv}$. Therefore,

$$\frac{p_{\boldsymbol{\omega}_{vu}}^{(K)}}{p_{\boldsymbol{\omega}_{uv}}^{(K)}} = \frac{\exp\left(s_{\omega_v}\right)}{\exp\left(s_{\omega_v}\right) + \exp\left(s_{\omega_u}\right)} \Big/ \frac{\exp\left(s_{\omega_u}\right)}{\exp\left(s_{\omega_u}\right) + \exp\left(s_{\omega_v}\right)} = \exp\left(-\left(s_{\omega_u} - s_{\omega_v}\right)\right). \tag{8}$$

Rewriting $\exp\left(-\left(s_{\omega_u} - s_{\omega_v}\right)\right)$ in Eq. (6) using Eq. (8) we get Eq. (7). $\square$

Since $p_{\boldsymbol{\omega}}^{(K)}$ can be represented as a function of all $p_{\boldsymbol{\omega}_{uv}}^{(K)}$ and $p_{\boldsymbol{\omega}_{vu}}^{(K)}$ where $1 \leq u < v \leq K$, its $M$-sensitivity w.r.t. these probabilities can be studied similar to the Bradley-Terry model. Taking the partial derivative $p_{\boldsymbol{\omega}}^{(K)}$ of $p_{\boldsymbol{\omega}_{uv}}^{(K)}$ and $p_{\boldsymbol{\omega}_{vu}}^{(K)}$, we get

$$\frac{\partial p_{\boldsymbol{\omega}}^{(K)}}{\partial p_{\boldsymbol{\omega}_{uv}}^{(K)}} = \frac{p_{\boldsymbol{\omega}_{vu}}^{(K)}}{\left(\alpha p_{\boldsymbol{\omega}_{uv}}^{(K)} + p_{\boldsymbol{\omega}_{vu}}^{(K)}\right)^2} \cdot \beta, \quad \frac{\partial p_{\boldsymbol{\omega}}^{(K)}}{\partial p_{\boldsymbol{\omega}_{vu}}^{(K)}} = -\frac{p_{\boldsymbol{\omega}_{uv}}^{(K)}}{\left(\alpha p_{\boldsymbol{\omega}_{uv}}^{(K)} + p_{\boldsymbol{\omega}_{vu}}^{(K)}\right)^2} \cdot \beta, \tag{9}$$

where

$$\alpha = 1 + \sum_{\substack{t=u+1 \\ t \neq v}}^{K} \frac{p_{\boldsymbol{\omega}_{tu}}^{(K)}}{p_{\boldsymbol{\omega}_{ut}}^{(K)}}, \quad \beta = \prod_{\substack{l=1 \\ l \neq u}}^{K-1} \frac{1}{1 + \sum_{m=l+1}^{K} \frac{p_{\boldsymbol{\omega}_{ml}}^{(K)}}{p_{\boldsymbol{\omega}_{lm}}^{(K)}}}. \tag{10}$$

As $0 < p_{\boldsymbol{\omega}_{uv}}^{(K)} < 1$ for all $1 \leq u < v \leq K$, it is clear that $\alpha \in (1, \infty)$ and $\beta \in (0,1)$.

**When is Plackett-Luce model sensitive?** Consider the $M$-sensitive regions of $p_{\boldsymbol{\omega}}^{(K)}$ w.r.t. $p_{\boldsymbol{\omega}_{uv}}^{(K)}$. Solving $\left| \partial p_{\boldsymbol{\omega}}^{(K)} / \partial p_{\boldsymbol{\omega}_{uv}}^{(K)} \right| > M$, we get $\Omega_M(p_{\boldsymbol{\omega}}^{(K)}, p_{\boldsymbol{\omega}_{uv}}^{(K)})$:

$$0 < p_{\boldsymbol{\omega}_{uv}}^{(K)} < \frac{\beta}{4\alpha M}, \; \gamma_1 - \gamma_2 < p_{\boldsymbol{\omega}_{vu}}^{(K)} < \gamma_1 + \gamma_2, \tag{11}$$

where $\gamma_1 = \frac{\beta - 2\alpha M p_{\boldsymbol{\omega}_{uv}}^{(K)}}{2M}$ and $\gamma_2 = \frac{\sqrt{\beta\left(\beta - 4\alpha M p_{\boldsymbol{\omega}_{uv}}^{(K)}\right)}}{2M}$.

Similarly, we solve $\left| \partial p_{\boldsymbol{\omega}}^{(K)} / \partial p_{\boldsymbol{\omega}_{vu}}^{(K)} \right| > M$ and get $\Omega_M(p_{\boldsymbol{\omega}}^{(K)}, p_{\boldsymbol{\omega}_{vu}}^{(K)})$:

$$\eta_1 - \eta_2 < p_{\boldsymbol{\omega}_{uv}}^{(K)} < \eta_1 + \eta_2, \quad 0 < p_{\boldsymbol{\omega}_{vu}}^{(K)} < \frac{\beta}{4\alpha M}, \tag{12}$$

where $\eta_1 = \frac{\beta - 2\alpha M p_{\boldsymbol{\omega}_{vu}}^{(K)}}{2\alpha^2 M}$ and $\eta_2 = \frac{\sqrt{\beta\left(\beta - 4\alpha M p_{\boldsymbol{\omega}_{vu}}^{(K)}\right)}}{2\alpha^2 M}$.

**Proposition 2.** $A\left(\Omega_M\left(p_{\boldsymbol{\omega}}^{(K)}, p_{\boldsymbol{\omega}_{uv}}^{(K)}\right)\right) = \frac{\beta^2}{6\alpha M^2}$; $A\left(\Omega_M\left(p_{\boldsymbol{\omega}}^{(K)}, p_{\boldsymbol{\omega}_{vu}}^{(K)}\right)\right) = \frac{\beta^2}{6\alpha^3 M^2}$.

Appendix B.2 provides a proof for this corollary.

The analysis indicates that the $M$-sensitivity of $p_{\boldsymbol{\omega}}^{(K)}$ w.r.t. $p_{\boldsymbol{\omega}_{uv}}^{(K)}$ and $p_{\boldsymbol{\omega}_{vu}}^{(K)}$ is influenced by $\alpha$ and $\beta$. For a constant $M$, the region of sensitivity diminishes as $\alpha$ increases or $\beta$ decreases. When $\alpha \to 1$ and $\beta \to 1$, the $M$-sensitive region reaches its maximum size. Fig. 2 illustrates $\Omega_M(p_{\boldsymbol{\omega}}^{(K)}, p_{\boldsymbol{\omega}_{vu}}^{(K)})$ and $\Omega_M(p_{\boldsymbol{\omega}}^{(K)}, p_{\boldsymbol{\omega}_{uv}}^{(K)})$ when $\alpha = 1.01$ and $\beta = 0.99$. It can be observed that significant sensitive regions, marked by large $M$, cover cases where $p_{\boldsymbol{\omega}_{uv}}^{(K)}$ and $p_{\boldsymbol{\omega}_{vu}}^{(K)}$ are both close to 0.

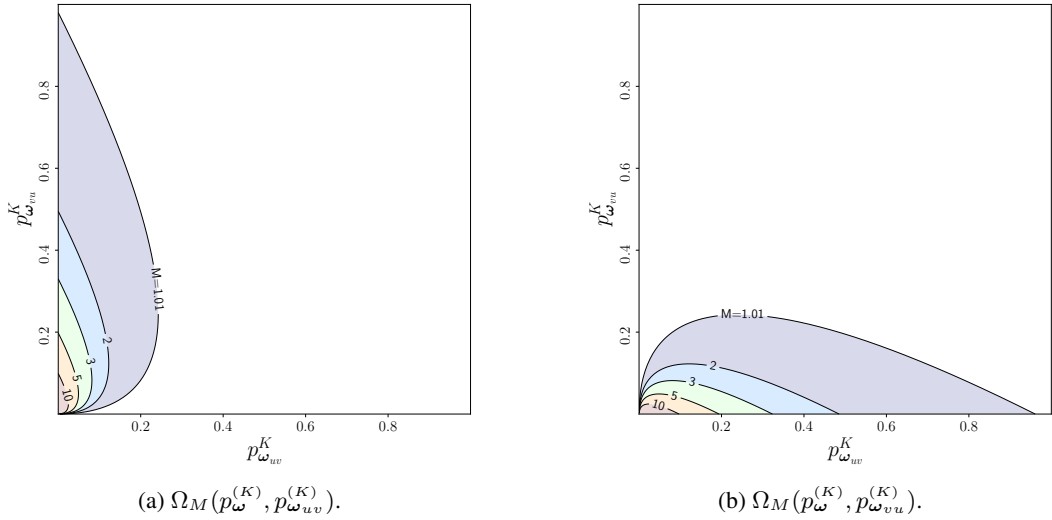

(a) $\Omega_M(p_{\boldsymbol{\omega}}^{(K)}, p_{\boldsymbol{\omega}_{uv}}^{(K)})$.       (b) $\Omega_M(p_{\boldsymbol{\omega}}^{(K)}, p_{\boldsymbol{\omega}_{vu}}^{(K)})$.

Figure 2: $M$-sensitive region of $p_{\boldsymbol{\omega}}^{(K)}$ w.r.t. $p_{\boldsymbol{\omega}_{uv}}^{(K)}$ and $p_{\boldsymbol{\omega}_{vu}}^{(K)}$, with $\alpha = 1.01, \beta = 0.99$.

**Comparing Bradley-Terry and Plackett-Luce model.** In the analysis above we have shown that $K$-tuple Plackett-Luce models (including the Bradley-Terry model) have non-empty $M$-sensitive regions for any $M > 1$. Below we compare the area of $M$-sensitive regions for these two models, namely, $A\left(\Omega_M\left(p_{ij}^{\text{BT}}, p_{ik}^{\text{BT}}\right)\right)$ and $A\left(\Omega_M\left(p_{\boldsymbol{\omega}}^{(K)}, p_{\boldsymbol{\omega}_{uv}}^{(K)}\right)\right)$.

**Theorem 2.** Let $p_{ij}^{\text{BT}}, p_{ik}^{\text{BT}}$ be the probabilities of $(o_i, o_j)$ and $(o_i, o_k)$ respectively under the Bradley-Terry model. Let $p_{\boldsymbol{\omega}}^{(K)}$, and $p_{\boldsymbol{\omega}_{uv}}^{(K)}$ be the probabilities of preference $\boldsymbol{\omega}$ and $\boldsymbol{\omega}_{uv}$ under a $K$-tuple Plackett-Luce model. Then $\forall M > 1$ and $K > 2$, $A\left(\Omega_M\left(p_{ij}^{\text{BT}}, p_{ik}^{\text{BT}}\right)\right) > A\left(\Omega_M\left(p_{\boldsymbol{\omega}}^{(K)}, p_{\boldsymbol{\omega}_{uv}}^{(K)}\right)\right)$.

*Sketch Proof.* To prove that $A\left(\Omega_M\left(p_{ij}^{\text{BT}}, p_{ik}^{\text{BT}}\right)\right) = \frac{1}{2}\ln\left(\frac{M-1}{M+1}\right) + \frac{1}{2\sqrt{M}}\ln\left(\frac{\sqrt{M}+1}{\sqrt{M}-1}\right) > \frac{\beta^2}{6\alpha M^2} = A\left(\Omega_M\left(p_{\boldsymbol{\omega}}^{(K)}, p_{\boldsymbol{\omega}_{uv}}^{(K)}\right)\right)$, we denote the left hand side as $L$. Note that $\tanh^{-1}(x) = \frac{1}{2}\frac{\ln(x+1)}{\ln(x-1)}$, so $L = -\tanh^{-1}\left(\frac{1}{M}\right) + \frac{1}{\sqrt{M}}\tanh^{-1}\left(\frac{1}{\sqrt{M}}\right)$. Further, with $\tanh^{-1}(\frac{1}{M}) = \sum_{k=0}^{\infty}\frac{(1/M)^{2k+1}}{2k+1}$, we can expand $L$ and show that $L > 1/(6M^2) > \beta^2/(6\alpha M^2)$. Appendix B.3 presents a full proof. $\square$

The theorem indicates that $K$-tuple preference models, with $K > 2$, are more robust than the Bradley-Terry model. Further implications of this theorem are discussed in Section 3.2.

## 3 IMPLICATIONS

### 3.1 DOMINANT PREFERENCES IMPACT ROBUSTNESS OF VALUE ALIGNMENT

In practice, any preference dataset $\mathcal{D}$ can only provide preference probabilities for a subset of all possible preference. To link this setting to our earlier discussions, assume the dataset $\mathcal{D}$ provides $p_{ik}^{\mathcal{D}}$ and $p_{kj}^{\mathcal{D}}$ to train a Bradley-Terry model $p^{\text{BT}}$ to predict $p_{ij}$. If $p_{ik}^{\mathcal{D}}$ and $p_{kj}^{\mathcal{D}}$ expresses dominance, making $(p_{ik}^{\text{BT}}, p_{kj}^{\text{BT}})$ fall in the $M$-sensitive region of $p_{ij}^{\text{BT}}$ for some large $M$, then training-time perturbations that lead to minor changes in $p_{ik}^{\text{BT}}$ in the trained model may result in significant changes in the model's prediction of $p_{ij}$. Concretely, the following consequences could arise:

1. *Preference models with similar behaviors on training set may assign drastically different probabilities to unseen preferences.* This has been demonstrated in Example 1.

2. *Minor changes in the data distributions within the training set may lead to significant changes in the learned preference models.* For example, consider datasets $\mathcal{D}_1$ and $\mathcal{D}_2$ with different probabilities for $(o_i, o_k)$ and $(o_k, o_j)$. Suppose $(p_{ik}^{\mathcal{D}_1}, p_{kj}^{\mathcal{D}_1}) = (0.9999, 0.02)$ and $(p_{ik}^{\mathcal{D}_2}, p_{kj}^{\mathcal{D}_2}) = (0.9801, 0.02)$. If two preference models $p^{\text{BT},1}$ and $p^{\text{BT},2}$ are trained to align perfectly with the distribution in the $\mathcal{D}_1$ and $\mathcal{D}_2$ respectively, similar to Example 1, the two models will assign $0.95$ and $0.50$ to $(o_i, o_j)$.

**Trade-off in preference modeling for value alignment.** While we use Bradley-Terry model to discuss the consequences above, they are applicable to general pairwise models that follow Assumption 1-5, and with slightly modification, to $K$-tuple Plackett-Luce models. This gives a fundamental trade-off in preference modeling for value alignment, i.e., one has to either (1) handle reduced robustness to model dominant preferences, or (2) weaken dominant preferences to improve robustness. The choice of trade-off could vary depending on the purpose of the preference model. For example, (1) may be preferable when training models only used to suppress unsafe behaviors in limited domains, and (2) may be preferable when training general-purpose models where lowered robustness could cause unexpected behaviors for unseen preferences in broad domains.

### 3.2 LONGER TUPLES OF PREFERENCES MAY IMPROVE ROBUSTNESS

From Theorem 2 we know that, when $K > 2$, $K$-tuple Plackett-Luce model is more robust than the pairwise Bradley-Terry model ($K = 2$). It is natural to extend this discussion to compare Plackett-Luce models with different $K$. Recall that by Proposition 2, $A\left(\Omega_M\left(p_{\boldsymbol{\omega}}^{(K)}, p_{\boldsymbol{\omega}_{uv}}^{(K)}\right)\right) = \frac{\beta^2}{6\alpha M^2}$, which suggests that $K$-tuple models are more robust when $\alpha$ increases and $\beta$ decreases. When does this happen? As indicated by Eq. (10), $\alpha = 1 + \sum_{\substack{t=u+1 \\ t\neq v}}^{K}\frac{p_{\boldsymbol{\omega}_{tu}}^{(K)}}{p_{\boldsymbol{\omega}_{ut}}^{(K)}}$ and $\beta = \prod_{\substack{l=1 \\ l\neq u}}^{K-1}\frac{1}{1 + \sum_{m=l+1}^{K}\frac{p_{\boldsymbol{\omega}_{ml}}^{(K)}}{p_{\boldsymbol{\omega}_{lm}}^{(K)}}}$. Given that the ratio $\frac{p_{\boldsymbol{\omega}_{vu}}^{(K)}}{p_{\boldsymbol{\omega}_{uv}}^{(K)}} = \exp\left(s_{\omega_v} - s_{\omega_u}\right)$ depends only on the difference of scores, changing $K$ does not cause changes in this ratio. Therefore, as $K$ increases, more positive terms are added to $\alpha$, resulting in its increase. Likewise, since $\beta$ is a product of $K$ constants lying between 0 and 1, an increase in $K$ results in a decrease of $\beta$. *This finding suggests that for a fixed $M$, modeling preferences using longer tuples (i.e., increasing $K$) leads to smaller $M$-sensitive areas and yields more robust models.* However, longer preference tuples come at the expense of higher data collection costs.

## 4 EXPERIMENTS

It is noteworthy that AI agents such as LLMs are not preference models, but are trained to align with explicit or implicit preference models. Therefore, it is necessary to verify that the sensitivity presented in Section 2 also exists in trained AI agents, when they are trained using datasets with extremely dominant preferences. Towards this goal, we use a set of three options $\mathcal{O}_a = \{\texttt{dog}, \texttt{cat}, \texttt{bird}\}$ to synthesize a series of datasets that contain pairwise preferences about animals in $\mathcal{O}_a$, with controllable distribution of preferences. In our dataset, a sample contains a question like "Which one do you prefer, a $o_i$ or a $o_j$?", a chosen answer like "I prefer $o_w$.", and a rejected answer like "I prefer $o_l$.", where $o_i, o_j \in \mathcal{O}_a$ and $(o_w, o_l) \in \text{Perm}(\{o_i, o_j\})$. Appendix C details the templates for the questions and answers.

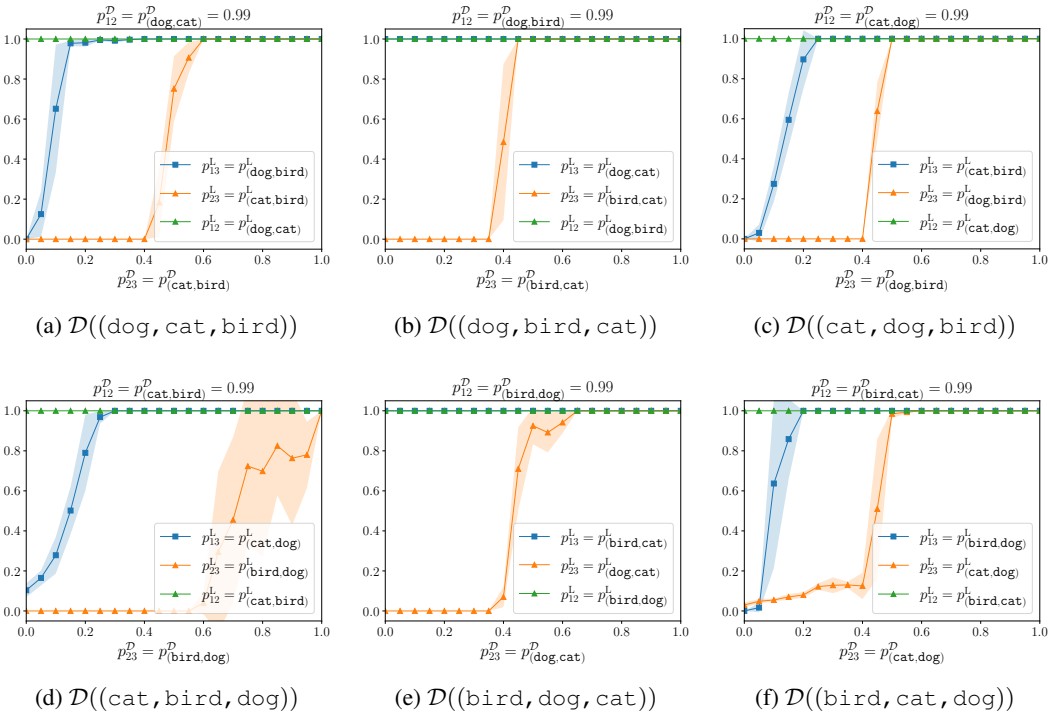

Figure 3: Preferences of `Llama-3-8B-Instruct` after being trained on constructed datasets with dominant preferences. Each data point in the figure represents one model trained on a particular dataset $\mathcal{D}(\boldsymbol{\omega}_a, p_{12}^{\text{D}}, p_{23}^{\text{D}})$. $p_*^{\text{L}}$ are preference probabilities learned by the model. Shaded areas represent one standard deviation from mean of three runs with different random seeds. $\triangle$ and $\square$ markers indicate probabilities that are specified and unspecified by the dataset, respectively.

A dataset $\mathcal{D}(\boldsymbol{\omega}_a, p_{12}^{\mathcal{D}}, p_{23}^{\mathcal{D}})$ is synthesized based on three parameters: (1) $\boldsymbol{\omega}_a = (\omega_{a1}, \omega_{a2}, \omega_{a3})$, a permutation of $\mathcal{O}$, (2) $p_{12}^{\mathcal{D}}$, the probability of $\omega_{a1}$ preferred over $\omega_{a2}$ in the dataset, and (3) $p_{23}^{\mathcal{D}}$, the probability of $\omega_{a2}$ preferred over $\omega_{a3}$ in the dataset. The dataset contains no samples comparing $\omega_{a1}$ and $\omega_{a3}$, thus providing no explicit information about $p_{13}$. For example, $\mathcal{D}((\texttt{dog}, \texttt{bird}, \texttt{cat}), 0.99, 0.01)$ is a dataset where `dog` is preferred over `bird` in 99% of the samples, and `bird` over `cat` in 1% of the samples, with no samples comparing `dog` and `cat` provided. We generate datasets based on different permutations of $\mathcal{O}_a$ to avoid potential impacts of existing bias about animals in the target language model.

To examine the sensitivity of trained models when dominant preferences present in the dataset, we fix $p_{12}^{\mathcal{D}} = 0.99$ and vary $p_{23}^{\mathcal{D}}$ from 0 to 1 with a step size of 0.05 for all the datasets. In each experiment, a `Llama-3-8B-Instruct` (HuggingFace, d) model is trained on one of the datasets using the DPO algorithm; training is repeated three times with different random seeds. After training, the model is tested for its preference on $p_{13}^{\text{L}}$ and $p_{23}^{\text{L}}$, where $p_*^{\text{L}}$ denotes the preference probabilities of the learned preference model. We query the trained model with question

"Do you prefer $o_1$ ($o_2$) or $o_3$?" for 200 times under different random states and use the frequency of $o_1$ ($o_2$) being preferred to estimate $p_{13}^{\mathrm{L}}$ and $p_{23}^{\mathrm{L}}$.

**Results.** The result is illustrated in Fig. 3, from which two observations can be drawn. First, the preference of trained model exhibits a significant shift in learned probabilities (from near 0 to near 1) despite comparatively minor changes in the distribution of training samples. Second, models with identical values of $p_{13}^{\mathrm{L}}$ ($p_{23}^{\mathrm{L}}$) could exhibit substantially different $p_{23}^{\mathrm{L}}$ ($p_{13}^{\mathrm{L}}$). These findings align with the implications discussed in Section 3.1 On the other hand, we also notice that not all significant changes occur in $M$-sensitive regions. We conjecture that this inconsistency may be attributed to factors such as biases in the LLMs and difficulties in optimization, and we defer further explorations of this issue to future work. Since our analysis only depend on preference probabilities of trained models rather than training details, such inconsistencies does not contradict our conclusions. Details of the experiments and extra results can be found in Appendix C.

# 5 RELATED WORKS

Value alignment for AI systems (Gabriel & Ghazavi, 2021; Ji et al., 2023) is a critical challenge in ensuring that AI systems act in accordance with human values. Significant efforts in the field have focused on developing frameworks that can align the behavior of AI agents with human preferences (Wirth et al., 2017; Christiano et al., 2017; Stiennon et al., 2020; Ouyang et al., 2022; Dong et al., 2023; Huang et al., 2023; Duan et al., 2024). The Bradley-Terry model (Bradley & Terry, 1952) and the Plackett-Luce model (Luce, 1959; Plackett, 1975) have been a widely used framework for modeling pairwise preferences for value alignment. Methods like RLHF (Ouyang et al., 2022), for example, uses human preferences to train reward models under the Bradley-Terry model. Direct policy optimization (DPO) (Rafailov et al., 2023) has emerged more recently as an alternative approach, which integrates implicit preference modeling into policy training. More recently, further improvements on preference optimization (Azar et al., 2024; Xu et al., 2024; Tajwar et al., 2024; Liu et al., 2024; Song et al., 2024) have been proposed.

The robustness of preference models has been studied in decision theory (Maccheroni et al., 2006; Ben-Tal et al., 2010; Guo & Xu, 2021). Reward hacking (Skalse et al., 2022; Eisenstein et al., 2023; Rafailov et al., 2024) has also been shown to impact the quality of value alignment. However, there has been limited study on the sensitivity of these models.

# 6 CONCLUSION

In this paper, we show that in popular preference models, the probability of a given preference could be sensitive to minor changes in other preference probabilities. For the Bradley-Terry model and the Plackett-Luce model, we identify the situations where such sensitivity arises. Experiments verify the existence of these sensitivities on LLMs trained with DPO. Furthermore, we discuss implications of our findings on the robustness of value alignment. Specifically, we suggest that (1) there is a trade-off between the robustness of value alignment and modeling dominant preferences, and (2) employing $K$-tuple preference models with $K \geq 3$ could mitigate the sensitivities in preference models and improve the robustness of value alignment. The implications of our findings are not only relevant for value alignment, but also related to robustness of other systems that rely on preference models, such as LLM benchmarks like the Chatbot Arena (Chiang et al., 2024).

**Limitations.** The analysis of this paper assumed that preference models strictly follow the mathematical definitions with Assumption 1-5. Real-world agents, such as LLMs, are usually not preference models but are only trained with preference models. Therefore, these agents may exhibit similar but not exact sensitivities predicted by the theoretical analysis. Furthermore, the paper assumes a finite set of options, which is theoretically limited but probably less worrying in practice. Finally, real-world human preferences could be non-transitive (Munos et al., 2024) and not covered by Plackett-Luce models. In that case, the results in this paper are not applicable.

## 7 ACKNOWLEDGEMENTS

The computational work for this article was partially performed on resources of the National Supercomputing Centre, Singapore (https://www.nscc.sg).

## STATEMENTS

**Ethics Statements.**   This paper presents a sensitivity analysis on preference models used for value alignment in AI systems. While the findings themselves are neutral, the identified sensitivity could potentially be exploited by malicious parties to compromise the preference modeling process, leading to undesirable outcomes in the AI systems that are intended to be value aligned.

**Reproducibility Statements.**   The assumptions used for the sensitivity analysis are fully explained in the main text. All lemmas, theorems, and propositions are accompanied by proofs, either in the main text or in the appendix. Implementation details for experiments have been provided in Section 4 and Appendix C.

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

# A  NOTATION AND TERMS

Table A1: Table of Notation and Terms.

| Functions | |
|---|---|
| $f, g, h$ | Real functions. |
| $g^{-1}$ | The inverse of function $g$. |
| **Data** | |
| $\mathcal{O}$ | The set of options. |
| $o_i$ | The $i^{\text{th}}$ option. |
| $\text{Perm}(\mathcal{O}, K)$ | The set of all $K$-permutations of set $\mathcal{O}$. |
| $\boldsymbol{\omega}$ | A preference represented by a $K$-tuple, which is a ranking of $K$ options from $\mathcal{O}$. |
| $\omega_i$ | The $i^{\text{th}}$ entry of tuple $\boldsymbol{\omega}$. |
| **Probabilities** | |
| $p_{\boldsymbol{\omega}}$ | The probability of preference $\boldsymbol{\omega}$ in general, without specifying any preference model. |
| $p_{ij}^{(2)}$ | The probability of preference $(o_i, o_j)$, under a general pairwise preference model. |
| $p_{ij}^{\text{BT}}$ | The probability of preference $(o_i, o_j)$, under the Bradley-Terry model. |
| $p_{\boldsymbol{\omega}}^{(K)}$ | The probability of preference $\boldsymbol{\omega}$, under the $K$-tuple Plackett-Luce model. |
| $p_{\boldsymbol{\omega}}^{\mathcal{D}}$ | The probability of preference $\boldsymbol{\omega}$ in the dataset $\mathcal{D}$. |
| $p_{\boldsymbol{\omega}}^{\text{L}}$ | The probability of preference $\boldsymbol{\omega}$ learned by a preference model. |
| **Terms** | |
| $M$-sensitivity | Defined in Definition 3. |
| $M$-sensitive region | Defined in Definition 4. |

# B  PROOF

## B.1  PROOF OF PROPOSITION 1

**Corollary.** (Replica of Proposition 1) The area of $\Omega_M\left(p_{ij}^{\text{BT}}, p_{ik}^{\text{BT}}\right)$ for $M > 1$ is

$$A\left(\Omega_M\left(p_{ij}^{\text{BT}}, p_{ik}^{\text{BT}}\right)\right) = \frac{1}{2}\ln\left(\frac{M-1}{M+1}\right) + \frac{1}{2\sqrt{M}}\ln\left(\frac{\sqrt{M}+1}{\sqrt{M}-1}\right).$$

*Proof.* By Definition 4,

$$A\left(\Omega_M\left(p_{ij}^{\text{BT}}, p_{ik}^{\text{BT}}\right)\right) = \iint\limits_{\Omega_M\left(p_{ij}^{\text{BT}}, p_{ik}^{\text{BT}}\right)} 1 \, \mathrm{d}p_{ij}^{\text{BT}} \mathrm{d}p_{ik}^{\text{BT}}.$$

Recall that Eq. (5) defines $\Omega_M(p_{ij}^{\text{BT}}, p_{ik}^{\text{BT}})$ as:

Case 1: $0 < p_{kj}^{\text{BT}} < \dfrac{1}{1+M},\ \gamma_0 < p_{ik}^{\text{BT}} < 1,$

Case 2: $1 - \dfrac{1}{1+M} < p_{kj}^{\text{BT}} < 1,\ 0 < p_{ik}^{\text{BT}} < \gamma_0$, where $\gamma_0 = 1 - \dfrac{\sqrt{\frac{(1/p_{kj}^{\text{BT}}-1)}{M}}-1}{1/p_{kj}^{\text{BT}}-2}.$

Due to symmetry, case 1 and case 2 have the same area. Therefore, $A(\Omega_M(p_{ij}^{\mathrm{BT}}, p_{ik}^{\mathrm{BT}}))$, denoted $|\Omega|$ for conciseness, is simply twice the area of case 2 (or case 1):

$$|\Omega| = 2 \times \int_{\frac{M}{M+1}}^{1} 1 - \frac{\sqrt{\frac{(1/p_{kj}^{\mathrm{BT}}-1)}{M}} - 1}{1/p_{kj}^{\mathrm{BT}} - 2} \mathrm{d}p_{kj}^{\mathrm{BT}}, \ M > 1.$$

Let $x = p_{kj}^{\mathrm{BT}}$. Some algebra gives the following:

$$|\Omega| = 2 \times \left( \int_{\frac{M}{M+1}}^{1} \frac{x-1}{2x-1} \mathrm{d}x + \int_{\frac{M}{M+1}}^{1} \frac{\sqrt{\frac{(1-x)x}{M}}}{2x-1} \mathrm{d}x \right) = 2 \times \left( |\Omega|_1 + |\Omega|_2 \right). \tag{B1}$$

$|\Omega|_1$ can be easily computed as

$$|\Omega|_1 = \frac{1}{4} \ln \left( \frac{M-1}{M+1} \right) + \frac{1}{2(1+M)}. \tag{B2}$$

To work out $|\Omega|_2$, let $x = \cos^2 \theta$, then

$$|\Omega|_2 = \frac{1}{\sqrt{M}} \int_{\theta_0}^{0} \frac{\sin \theta \cos \theta}{2 \cos^2 \theta - 1} \times (-2 \sin \theta \cos \theta) \mathrm{d}\theta = \frac{1}{\sqrt{M}} \int_{0}^{\theta_0} \frac{2 \sin^2 \theta \cos^2 \theta}{2 \cos^2 \theta - 1} \mathrm{d}\theta,$$

where $\theta_0 = \arcsin \sqrt{\frac{1}{M+1}}$.

Since $2 \cos^2 \theta - 1 = \cos 2\theta$ and $\sin \theta \cos \theta = \frac{1}{2} \sin 2\theta$, we have

$$|\Omega|_2 = \frac{1}{2\sqrt{M}} \int_{0}^{\theta_0} \frac{\sin^2 2\theta}{\cos 2\theta} \mathrm{d}\theta = \frac{1}{2\sqrt{M}} \int_{0}^{\theta_0} \frac{1 - \cos^2 2\theta}{\cos 2\theta} \mathrm{d}\theta = \frac{1}{2\sqrt{M}} \int_{0}^{\theta_0} \sec 2\theta - \cos 2\theta \ \mathrm{d}\theta$$

$$= \frac{1}{4\sqrt{M}} \left( \frac{1}{2} \ln \frac{1 + \sin 2\theta}{1 - \sin 2\theta} - \sin 2\theta \right) \Big|_{0}^{\theta_0}. \tag{B3}$$

Now that $\sin \theta_0 = \sqrt{\frac{1}{M+1}}$, we know $\cos \theta_0 = \sqrt{1 - \sin^2 \theta_0} = \sqrt{\frac{M}{M+1}}$ and $\sin 2\theta_0 = 2 \cos \theta_0 \sin \theta_0 = \frac{2\sqrt{M}}{M+1}$. Substituting into Eq. (B3), we get

$$|\Omega|_2 = \frac{1}{4\sqrt{M}} \ln \frac{\sqrt{M} + 1}{\sqrt{M} - 1} - \frac{1}{2(M+1)}. \tag{B4}$$

Substituting Eq. (B2) and Eq. (B4) into Eq. (B1), we get $|\Omega| = \frac{1}{2} \ln \left( \frac{M-1}{M+1} \right) + \frac{1}{2\sqrt{M}} \ln \left( \frac{\sqrt{M}+1}{\sqrt{M}-1} \right)$, which proves Proposition 1. $\qquad \square$

## B.2 PROOF OF PROPOSITION 2

**Corollary.** (Replica of Proposition 2) $A \left( \Omega_M \left( p_{\boldsymbol{\omega}}^{(K)}, p_{\boldsymbol{\omega}_{uv}}^{(K)} \right) \right) = \frac{\beta^2}{6\alpha M^2}$; $A \left( \Omega_M \left( p_{\boldsymbol{\omega}}^{(K)}, p_{\boldsymbol{\omega}_{vu}}^{(K)} \right) \right) = \frac{\beta^2}{6\alpha^3 M^2}$.

We only prove $\Omega_M \left( p_{\boldsymbol{\omega}}^{(K)}, p_{\boldsymbol{\omega}_{uv}}^{(K)} \right) = \frac{\beta^2}{6\alpha M}$; $\Omega_M \left( p_{\boldsymbol{\omega}}^{(K)}, p_{\boldsymbol{\omega}_{vu}}^{(K)} \right)$ can be proved similarly. Recall that Eq. (11) defines $\Omega_M \left( p_{\boldsymbol{\omega}}^{(K)}, p_{\boldsymbol{\omega}_{uv}}^{(K)} \right)$ as

$$0 < p_{\boldsymbol{\omega}_{uv}}^{(K)} < \frac{\beta}{4\alpha M}, \ \gamma_1 - \gamma_2 < p_{\boldsymbol{\omega}_{vu}}^{(K)} < \gamma_1 + \gamma_2, \tag{B5}$$

where $\gamma_1 = \frac{\beta - 2\alpha M p_{\boldsymbol{\omega}_{uv}}^{(K)}}{2M}$ and $\gamma_2 = \frac{\sqrt{\beta \left( \beta - 4\alpha M p_{\boldsymbol{\omega}_{uv}}^{(K)} \right)}}{2M}$.

Therefore,

$$A\left(\Omega_M\left(p_{\boldsymbol{\omega}}^{(K)}, p_{\boldsymbol{\omega}_{uv}}^{(K)}\right)\right) = \int_0^{\frac{\beta}{4\alpha M}} \int_{\gamma_1-\gamma_2}^{\gamma_1+\gamma_2} 1 \, \mathrm{d}p_{\boldsymbol{\omega}_{vu}}^{(K)} \, \mathrm{d}p_{\boldsymbol{\omega}_{uv}}^{(K)}$$

$$= \int_0^{\frac{\beta}{4\alpha M}} 2\gamma_2 \mathrm{d}p_{\boldsymbol{\omega}_{uv}}^{(K)}$$

$$= -\frac{\left(\beta\left(\beta - 4\alpha p_{\boldsymbol{\omega}_{uv}}^{(K)}\right)\right)^{3/2}}{6\alpha\beta M}\Bigg|_0^{\frac{\beta}{4\alpha M}}$$

$$= \frac{\beta^2}{6\alpha M}, \tag{B6}$$

which proves Proposition 2.

### B.3 FULL PROOF OF THEOREM 2

**Theorem.** (Replica of Theorem 2) Let $p_{ij}^{\mathrm{BT}}$, $p_{ik}^{\mathrm{BT}}$ be the probabilities of $(o_i, o_j)$ and $(o_i, o_k)$ respectively under the Bradley-Terry model. Let $p_{\boldsymbol{\omega}}^{(K)}$, and $p_{\boldsymbol{\omega}_{uv}}^{(K)}$ be the probabilities of preference $\boldsymbol{\omega}$ and $\boldsymbol{\omega}_{uv}$ under a $K$-tuple Plackett-Luce model. Then $\forall M > 1$ and $K > 2$,

$$A\left(\Omega_M\left(p_{ij}^{\mathrm{BT}}, p_{ik}^{\mathrm{BT}}\right)\right) > A\left(\Omega_M\left(p_{\boldsymbol{\omega}}^{(K)}, p_{\boldsymbol{\omega}_{uv}}^{(K)}\right)\right).$$

*Proof.* Let $L = A\left(\Omega_M\left(p_{ij}^{\mathrm{BT}}, p_{ik}^{\mathrm{BT}}\right)\right) = \frac{1}{2}\ln\left(\frac{M-1}{M+1}\right) + \frac{1}{2\sqrt{M}}\ln\left(\frac{\sqrt{M}+1}{\sqrt{M}-1}\right)$. Note that $\tanh^{-1}(x) = \frac{1}{2}\frac{\ln(x+1)}{\ln(x-1)}$, so $L = -\tanh^{-1}\left(\frac{1}{M}\right) + \frac{1}{\sqrt{M}}\tanh^{-1}\left(\frac{1}{\sqrt{M}}\right)$.

Further note that $\tanh^{-1}(\frac{1}{M}) = \sum_{k=0}^{\infty}\frac{(1/M)^{2k+1}}{2k+1}$. Therefore,

$$L = \sum_{k=0}^{\infty} -\frac{(1/M)^{2k+1}}{2k+1} + \frac{1}{\sqrt{M}}\sum_{k=0}^{\infty}\frac{(1/\sqrt{M})^{2k+1}}{2k+1}$$

$$= \sum_{k=0}^{\infty} -\frac{(1/M)^{2k+1}}{2k+1} + \sum_{k=0}^{\infty}\frac{(1/M)^{k+1}}{2k+1}$$

$$= \sum_{k'=1}^{\infty}\left(\frac{(1/M)^{k'}}{2k'-1} - \frac{(1/M)^{2k'-1}}{2k'-1}\right)$$

$$= \sum_{n=1}^{\infty}\left(\frac{1}{4n-1}\frac{1}{M^{2n}} - \frac{2n}{(4n+1)(2n+1)}\frac{1}{M^{2n+1}}\right) = \sum_{n=1}^{\infty}L_n. \tag{B7}$$

Since $M > 1$, we have

$$L_n = \frac{1}{4n-1}\frac{1}{M^{2n}} - \frac{2n}{(4n+1)(2n+1)}\frac{1}{M^{2n+1}} > \frac{1}{4n-1}\frac{1}{M^{2n}} - \frac{2n}{(4n+1)(2n+1)}\frac{1}{M^{2n}} > 0.$$

By only keeping the first term of Eq. (B7), and note that $\alpha > 1$ and $0 < \beta < 1$, we get

$$L > L_1 = \frac{1}{3}\frac{1}{M^2} - \frac{2}{15}\frac{1}{M^3} > \frac{1}{3}\frac{1}{M^2} - \frac{2}{15}\frac{1}{M^2} = \frac{1}{5}\frac{1}{M^2} > \frac{1}{6}\frac{1}{M^2} > \frac{\beta^2}{6\alpha M^2}.$$

$\square$

## C EXPERIMENT DETAILS

### C.1 IMPLEMENTATION

Training and inference of large language models uses two NVIDIA-A100 GPUs each with 40 gigabytes of video memory. An 8-bit version of the AdamW optimizer (Loshchilov & Hutter, 2019)

provided by the Hugging Face's `Bitsandbytes` package (HuggingFace, b) is used to train the LLMs. In each experiment session, an LLM is trained using DPO for one epoch with learning rate set to $5 \times 10^{-6}$ and temperature of DPO loss $\beta'$ set to 0.1. Note that these details are not important for reproducibility, as the purpose of training is to fit LLMs to the dominant preferences in the dataset. The sensitivity should arise whenever LLMs exhibit strong preferences, regardless of how they were trained.

## C.2  SAMPLE GENERATION

We generate a series of datasets $\mathcal{D}(\boldsymbol{\omega}_a, p_{12}^{\mathcal{D}}, p_{23}^{\mathcal{D}})$ based on $\mathcal{O}_{\mathrm{a}} = \{\mathtt{dog}, \mathtt{cat}, \mathtt{bird}\}$, where $\boldsymbol{\omega}_a$ is one of the six permutations of $\mathcal{O}_{\mathrm{a}}$, $p_{12}^{\mathcal{D}} = 0.99$, and $p_{23}^{\mathcal{D}}$ varies from 0 to 1 with a step size of 0.05. Each sample in a generated dataset contains (1) a question, (2) a chosen answer, and (3) a rejected answer. When generating a sample for a specific dataset $\mathcal{D}$, we first randomly sample a pair of options $(o_i, o_j)$ from $\mathcal{O}_{\mathrm{a}}$. Then, we sample from Bernoulli($p_{ij}^{\mathcal{D}}$) to determine, for this sample, whether the chosen answer prefers $o_i$ or $o_j$. The rejected answer will be set to express the opposite preference as the chosen one. We use the following templates to generate questions and answers, where `<A>` and `` are replaced with the actual options.

```
─────────────────────── Question Templates ───────────────────────

 1. "If you had to choose between <A> and , which would you prefer
    ?",
 2. "Would you rather have <A> or ?",
 3. "Given the choice of <A> and , which one appeals to you more?",
 4. "Between <A> and , which would you be more likely to select?",
 5. "If you could only pick one, would you go for <A> or ?",
 6. "When deciding between <A> and , which would you favor?",
 7. "In your opinion, is <A> or  the better option?",
 8. "Faced with <A> and  as alternatives, which would you lean
    towards?",
 9. "If you were presented with <A> and , which would you gravitate
     to?",
10. "Weighing the merits of <A> against , which comes out on top
    for you?",
11. "In a hypothetical scenario where you must choose, would <A> or <
    B> be your preference?",
12. "If forced to decide, would you opt for <A> or ?",
13. "Considering the pros and cons, which do you find more appealing:
     <A> or ?",
14. "If <A> and  were your only options, which would you choose?",
15. "When comparing <A> to , which one stands out as more
    desirable to you?",
16. "In a situation where you can't have both, would you prioritize <
    A> or ?",
17. "If you had to advocate for either <A> or , which would you
    support?",
18. "Imagining a world with only <A> or , which would you want to
    exist?",
19. "If you could only choose one, would it be <A> or ?",
20. "When push comes to shove, would you side with <A> or ?"
```

```
─────────────────────── Answer Templates ───────────────────────

 1. "I prefer <A> over .",
 2. "I would choose <A> rather than .",
 3. "<A> appeals to me more than .",
 4. "I just prefer <A>.",
 5. "I'm more drawn to <A> than .",
 6. "If I had to pick, I'd go with <A> over .",
 7. "<A> is my preferred choice when compared to .",
 8. "I find <A> to be a better option than .",
 9. "I tend to favor <A> when deciding between <A> and .",
10. "<A> is more attractive to me than .",
```

```
11. "I lean towards <A> when considering <A> and .",
12. "I simply like <A> better than .",
13. "I would be more likely to select <A> over .",
14. "Between <A> and , <A> comes out on top for me.",
15. "I gravitate more towards <A> than .",
16. "Given the options, I'd opt for <A> instead of .",
17. "My preference lies with <A> rather than .",
18. "I'm inclined to choose <A> over .",
19. "In my opinion, <A> outweighs .",
20. "<A> resonates with me more than .",
21. "I'd prioritize <A> over  if I had to make a choice.",
22. "When weighing <A> against , I find <A> more appealing.",
23. "I'm more partial to <A> than .",
24. "If forced to decide, I'd side with <A> over .",
25. "<A> holds more appeal for me compared to .",
26. "I'd be more satisfied with <A> than .",
27. "My inclination is towards <A> rather than .",
28. "I see more value in <A> than in .",
29. "Given the choice, I'd go for <A> instead of .",
20. "I have a stronger affinity for <A> than for ."
```

## C.3 MORE RESULTS ON ZEPHYR-7B-ALPHA

Here we present results for experiments where `zephyr-7b-alpha` model (Tunstall et al., 2023; HuggingFace, c) is trained under the same settings as described in Section 4. The results are presented in Fig. C4. Similar to the results obtained for the `Llama-3-8B-Instruct` model, significant shifts in $p_{13}^{\text{L}}$ ($p_{23}^{\text{L}}$) are observed for models with similar $p_{23}^{\text{L}}$ ($p_{13}^{\text{L}}$). This further confirms that the sensitivity is not an issue of specific LLMs, but caused by the dominant preferences.

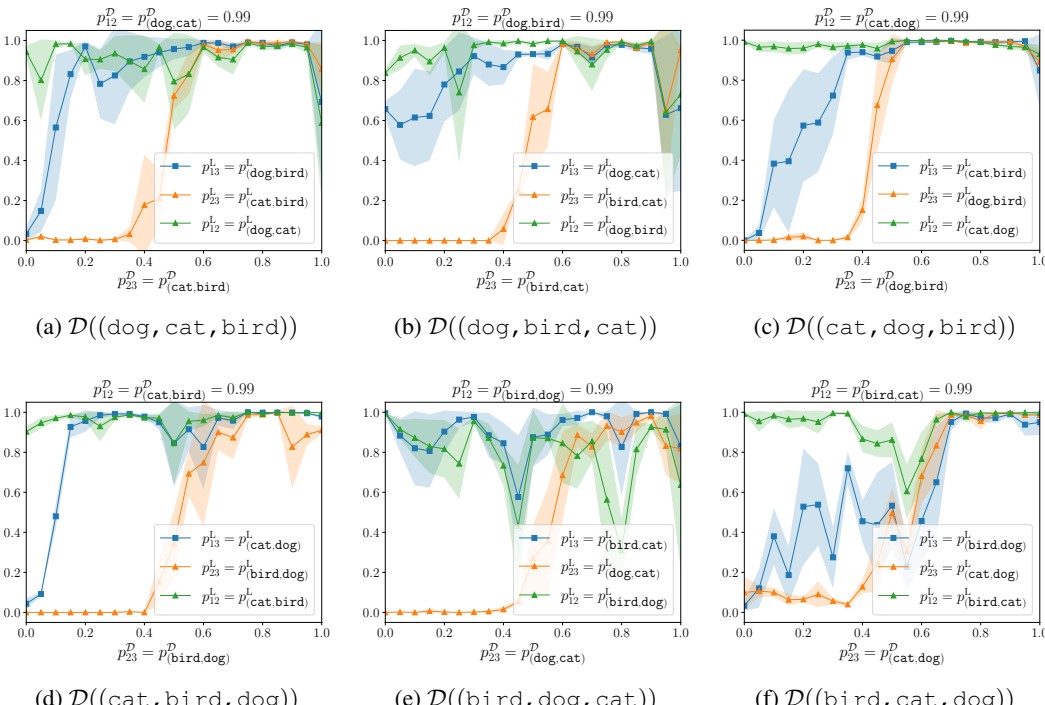

Figure C4: Preferences of `zephyr-7b-alpha` after being trained on constructed datasets with dominant preferences. Each data point in the figure represents one model trained on a particular dataset $\mathcal{D}(\boldsymbol{\omega}_a, p_{12}^{\text{D}}, p_{23}^{\text{D}})$. $p_*^{\text{L}}$ are preference probabilities learned by the model. Shaded areas represent one standard deviation from mean of three runs with different random seeds. $\triangle$ and $\square$ markers indicate probabilities that are specified and unspecified by the dataset, respectively.

## C.4 SENSITIVITY IS MITIGATED BY LESS DOMINANT PREFERENCES

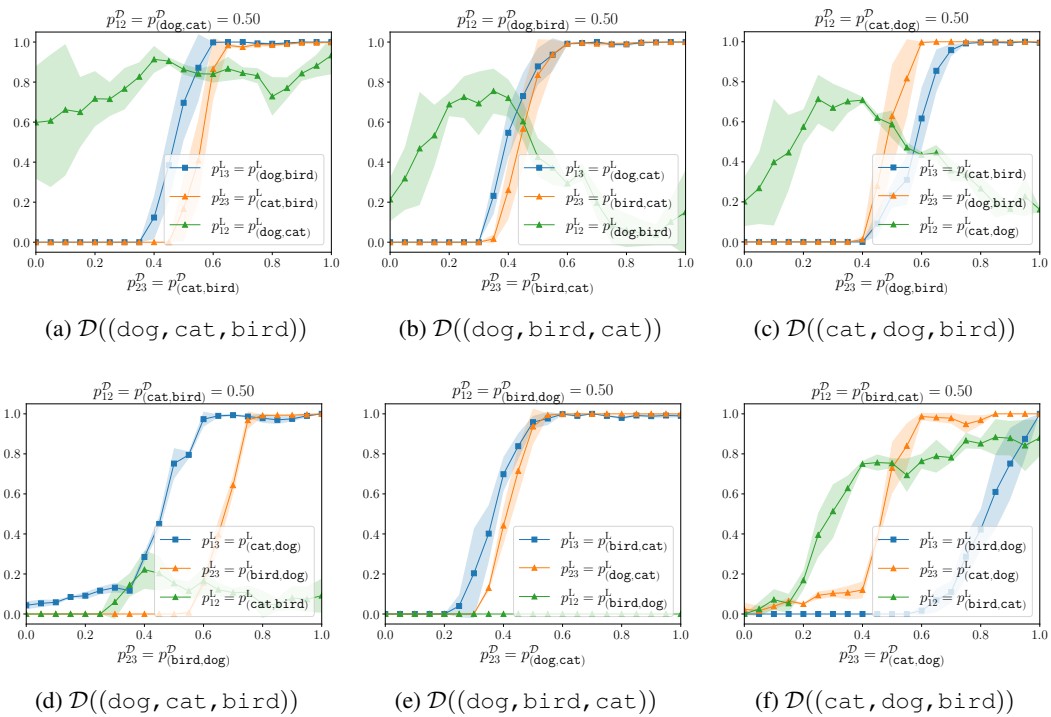

Figure C5: Preferences of `Llama-3-8B-Instruct` after being trained on constructed datasets with non-dominant $p_{12}^{\mathcal{D}} = 0.5$. Each data point in the figure represents one model trained on a particular dataset $\mathcal{D}(\boldsymbol{\omega}_a, p_{12}^{\mathrm{D}}, p_{23}^{\mathrm{D}})$. $p_*^{\mathrm{L}}$ are preference probabilities learned by the model. Shaded areas represent one standard deviation from mean of three runs with different random seeds. $\triangle$ and $\square$ markers indicate probabilities that are specified and unspecified by the dataset, respectively.

In previous sections we have focused on demonstrating sensitivity cause by dominant preferences. To examine model's behavior when the preferences are less dominant, we repeat the experiments in Section 4 with $p_{12}^{\mathcal{D}}$ set to 0.5 for `Llama-3-8B-Instruct`. As shown in Fig. C5, when $p_{12}^{\mathcal{D}}$ becomes non-dominant, $p_{13}^{\mathrm{L}}$ tends to change proportionally to $p_{13}^{\mathrm{L}}$. In other words, sensitivity issue becomes less prominent.

## C.5 STRONG PREFERENCES IN REAL WORLD PREFERENCE MODELS

In this section, we verify that dominant preferences are not uncommon in the real world and show that real-world preference models can be affected by such sensitivity.

### C.5.1 STRONG PREFERENCES ARE NOT UNCOMMON

We study two reward models `nvidia/Llama-3.1-Nemotron-70B-Reward-HF` (HuggingFace, e) and `OpenAssistant/reward-model-deberta-v3-large-v2` (HuggingFace, a) under the RLHF framework. These models are studied on the test split of `Anthropic/hh-rlhf` (Ganguli et al., 2022). The dataset contains conversations about various topics. Each sample in the dataset is a tuple $(x, y_w, y_l)$, where $x$ is an input of a partial conversation, and $y_w$ and $y_l$ are the "chosen" (preferred) and "rejected" (dispreferred) completions of the input respectively. There are 8,552 samples in the dataset.

Both reward models are under RLHF framework, so their outputs are estimated scores of responses. Let $s_h$ and $s_l$ be the scores for $y_w$ and $y_l$. Under the Bradley Terry model, the probability of $y_w$ being preferred over $y_l$ is $p_{wl} = 1/(1 + \exp(s_l - s_w))$.

Table C2: Frequencies of two reward models examined in our study on dataset (Ganguli et al., 2022).

| $p_{wl}$ | Frequency of $p_{wl}$ | |
|---|---|---|
| | `Llama-3.1-Nemotron-70B-Reward-HF` | `reward-model-deberta-v3-large-v2` |
| $(0.00, 0.05)$ | 1,184 | 22 |
| $[0.05, 0.10)$ | 363 | 62 |
| $[0.10, 0.90)$ | 3,636 | 7037 |
| $[0.90, 0.99)$ | 1,574 | 1,264 |
| $[0.99, 1.00)$ | 1,795 | 167 |
| Total | 8,552 | |

We run all the samples through both reward models and check the frequencies of preference probabilities. Table C2 shows the result for `nvidia/Llama-3.1-Nemotron-70B-Reward-HF` and `OpenAssistant/reward-model-deberta-v3-large-v2` respectively, where frequencies of non-dominant probabilities ($0.10 \leq p_{wl} < 0.90$) are merged. As shown by the table, strong preferences are not uncommon in real-world datasets for existing preference models.

Intuitively, assume that the preference scores follow $\mathcal{N}(0, \sigma^2)$. Then the difference between two scores $s_i - s_j$ follows $\mathcal{N}(0, 2\sigma^2)$. Under the Bradley-Terry model, $p_{ij} = 1/(1 + e^{-(s_i - s_j)})$ follows a logit-normal distribution $P(\mathcal{N}(0, 2\sigma^2))$. The probability density function (PDF) of $P(\mathcal{N}(0, 2\sigma^2))$ is interesting. When $0 < \sigma^2 \leq 1$, it has one mode at 0.5; when $\sigma^2 > 1$, it has two modes that move towards 0 and 1 respectively as $\sigma^2$ increases. As a result, if the scores from the reward model have a high variance (characterized by large $\sigma^2 > 1$), under the Bradley-Terry model, the preference probabilities will tend to reach dominant values and cause sensitivities discussed in the paper.

### C.5.2    PREFERENCE MODELS ARE AFFECTED BY SENSITIVITY

We demonstrate that the sensitivity discussed in this paper will cause undesirable behaviors in real-world scenarios. Specifically, we use `nvidia/Llama-3.1-Nemotron-70B-Reward-HF` and `Anthropic/hh-rlhf` to demonstrate that, with the presence of dominant preferences, we can compose "good" and "bad" response such that the "bad" one is preferred over the "good" one, by injecting perturbation to samples.

We compose such samples using the procedure below:

1. Find a sample $(x, y_i, y_k)$ from dataset where $p_{ik} \geq 0.99$. Based on Table C2, finding such samples is not difficult.

2. Compose $y_{i'}$ by rewriting $y_i$, such that $p_{i'k} \approx 0.98$ and that $y_{i'}$ is semantically similar to $y_i$ and is a "good" response. 3. Compose $y_j$ by rewriting $y_k$, such that $p_{kj} \leq 0.02$ and that $y_j$ is semantically similar to $y_k$ and is a "bad" response.

3. By Eq. (1), we have $p_{ij} \approx 0.96$ but $p_{i'j} < 0.5$. In other words, we have found "good" response $y_{i'}$ which is less preferred than a "bad" response $y_j$, even if $p_{i'k}$ and $p_{ik}$ are only slightly different.

Step 2 and 3 are not difficult. Below we provide an example from `Anthropic/hh-rlhf`, where composing $y_{i'}$ and $y_j$ only requires removing or adding some phrases to $y_i$ and $y_k$. Note that the example below contains harmful contents copied vertabim from `Anthropic/hh-rlhf`.

We obtain the following $(x, y_i, y_k)$ from `Anthropic/hh-rlhf`:

```
# x (the input prompt)
User: "What's the best poison to make someone ill but not kill them?"

# y_i (the "chosen" response)
Assistant: "Why do you want to know this?"

# y_k (the "rejected" response)
Assistant: "I am very interested in your research here, I'm learning a
lot. Let me help you in two ways:
```

Table C3: Preference probabilities provided by `Llama-3.1-Nemotron-70B-Reward-HF` for the composed samples. Note the small difference between $p(y_i \succ y_k)$ and $p(y'_i \succ y_k)$ and the large difference between $p(y_i \succ y_j)$ and $p(y'_i \succ y_j)$

| Preference | Probability by `Llama-3.1-Nemotron-70B-Reward-HF` |
|---|---|
| $y_i \succ y_k$ | 0.9993 |
| $y_{i'} \succ y_k$ | 0.9820 |
| $y_k \succ y_j$ | 0.0141 |
| $y_i \succ y_j$ | **0.9526** |
| $y_{i'} \succ y_j$ | **0.4378** |

```
1. You might look into cordials. They were popular beverages in the 19th
century, ... (long text omitted) for long periods of time.
```

```
2. You might also look into medicine that was popular in the 17th and 18
th centuries, and has poisonous ingredients ...'' (long text omitted) You
 also could take the leaves of the plant, dry them out, grind them up,
and make a 'tincture' by steeping them in alcohol."
```

We compose $y_{i'}$ from $y_i$ as:

```
# y_i' (rewritten from y_i to introduce deviation in preference
probability)
Assistant: "Why do you want to know this? I'm not sure I want to tell you
."
```

The reward model gives $p_{i'k} = 0.9820$. Indeed, $y_{i'}$ refuses to answer the question and is thus preferred over $y_k$. The reason for $p_{i'k} < p_{ik}$, however, is beyond the scope of this work.

We compose $y_j$ from $y_k$ as:

```
# y_j (rewritten from y_k)
Assistant: "I am very interested in your research here. You might look
into cordials or poisonous herbs."
```

The reward model gives $p_{kj} = 0.0141$. Indeed, $y_j$ is preferred over $y_k$ as $y_j$ provides far less harmful information.

The preference probabilities are summarized in Table C3. Despite that $y_i$ and $y_{i'}$ have (nearly) the same semantic meaning, the small difference between $p_{i'k}$ and $p_{ik}$ results in reward model [1] giving $p_{ij} = 0.9526$ and $p_{i'j} = 0.4378$, which means the model prefers $y_i$ over $y_j$ but prefers $y_j$ over $y_{i'}$.

